# The Multifaceted Roles of NRF2 in Cancer: Friend or Foe?

**DOI:** 10.3390/antiox13010070

**Published:** 2024-01-02

**Authors:** Christophe Glorieux, Cinthya Enríquez, Constanza González, Gabriela Aguirre-Martínez, Pedro Buc Calderon

**Affiliations:** 1State Key Laboratory of Oncology in South China, Collaborative Innovation Center for Cancer Medicine, Sun Yat-sen University Cancer Center, Guangzhou 510060, China; 2Química y Farmacia, Facultad de Ciencias de la Salud, Universidad Arturo Prat, Iquique 1100000, Chile; cenriquez@estudiantesunap.cl (C.E.); constanzagonzalez@estudiantesunap.cl (C.G.); gaguirre@unap.cl (G.A.-M.); 3Programa de Magister en Ciencias Químicas y Farmacéuticas, Facultad de Ciencias de la Salud, Universidad Arturo Prat, Iquique 1100000, Chile; 4Instituto de Química Medicinal, Universidad Arturo Prat, Iquique 1100000, Chile; 5Research Group in Metabolism and Nutrition, Louvain Drug Research Institute, Université Catholique de Louvain, 1200 Brussels, Belgium

**Keywords:** NRF2, cancer, metabolism, tumor immunology, phase separation, metastasis, LncRNA, NRF2 activators, NRF2 inhibitors, natural compounds

## Abstract

Physiological concentrations of reactive oxygen species (ROS) play vital roles in various normal cellular processes, whereas excessive ROS generation is central to disease pathogenesis. The nuclear factor erythroid 2-related factor 2 (NRF2) is a critical transcription factor that regulates the cellular antioxidant systems in response to oxidative stress by governing the expression of genes encoding antioxidant enzymes that shield cells from diverse oxidative alterations. NRF2 and its negative regulator Kelch-like ECH-associated protein 1 (KEAP1) have been the focus of numerous investigations in elucidating whether NRF2 suppresses tumor promotion or conversely exerts pro-oncogenic effects. NRF2 has been found to participate in various pathological processes, including dysregulated cell proliferation, metabolic remodeling, and resistance to apoptosis. Herein, this review article will examine the intriguing role of phase separation in activating the NRF2 transcriptional activity and explore the NRF2 dual impacts on tumor immunology, cancer stem cells, metastasis, and long non-coding RNAs (LncRNAs). Taken together, this review aims to discuss the NRF2 multifaceted roles in both cancer prevention and promotion while also addressing the advantages, disadvantages, and limitations associated with modulating NRF2 therapeutically in cancer treatment.

## 1. Introduction

Cancer is a complex ailment characterized by disrupted cell proliferation and various distinct features, including alterations in cellular metabolism, evasion of the immune system, and the ability to metastasize [1]. It is a leading cause of global mortality, with 19.3 million new cases and 9.9 million fatalities reported in 2020 [2]. Among all the cancer types, lung cancer ranks as the primary cause of death among men and the second leading cause among women, with breast cancer also being a significant disease with high mortality rates. The onset of lung cancer has been closely linked to tobacco smoking [3,4]. Given the substantial impact of cancer on public health worldwide, extensive efforts are being undertaken to advance therapeutic options and preventative measures in an attempt to cure the disease and reduce its incidence.

The regulation of reactive oxygen species (ROS) metabolism plays a vital role in numerous biological processes in both normal and cancer cells. Anomalies in the expression of antioxidant enzymes are frequently observed in cancer [5,6,7,8,9,10]. Moreover, mitochondrial dysfunction [11], and the activation of pro-oxidant enzymes, such as NADPH oxidases [12], contribute to increased ROS production in human cancer cells, trigger pro-tumorigenic signals, stimulate cell proliferation, and instigate DNA damage, resulting in genetic instability and mutations. Consequently, there has been a proposal to explore the use of pro-oxidant treatments to target the redox status of cancer cells, with the aim of enhancing the treatment outcomes of chemotherapy [13,14,15,16,17,18] and immunotherapy [19,20,21,22]. In this context, any increase in ROS levels poses a threat to the delicate redox equilibrium of cancer cells, rendering them susceptible to ROS-based therapy and promoting tumor cell death. Nevertheless, tumor cells can develop adaptive mechanisms in response to such heightened ROS levels by increasing their antioxidant capacity, thereby reducing ROS to non-toxic levels while preserving pro-tumorigenic signaling, promoting genetic instability, and resisting apoptosis [16].

The nuclear factor erythroid 2-related factor 2 (NRF2), originally identified by Venugopal and Jaiswal, plays a pivotal role in safeguarding cells against oxidative stress [23]. This transcription factor regulates the expression of numerous genes, including antioxidant enzymes responsible for shielding cells from various oxidative alterations. Owing to the dual nature of ROS in cancer, NRF2 and its antagonist, Kelch-like ECH-associated protein 1 (KEAP1), have become subjects of debate regarding their precise roles in either preventing or, conversely, promoting tumor progression [24]. In this review, we discuss the roles of NRF2 in cancer and the potential advantages and disadvantages associated with modulating its activity in cancer treatment.

## 2. The Transcription Factor NRF2

### 2.1. NRF Family Members

Vertebrates harbor three NRF factors. In contrast to NRF1, NRF2 does not possess an essential role in embryonic development. NRF1 knockout mice, for instance, succumb at approximately 17 days of gestation [25], whereas NRF2 knockout mice progress and thrive normally [26]. The mRNA expression of the third family member, NRF3, is abundant in the cornea, skin, and bladder, but is low in the major detoxification sites, notably the liver [27]. The NRF3 function remains undetermined [28,29,30,31], though recent propositions have implicated it in tumorigenesis and cancer malignancy [27,32].

NRF2 was initially isolated from a cDNA library derived from K562 cells to investigate distinct DNA-binding proteins [33]. The identified protein was named NRF2 due to its sequence resemblance to the nuclear factor erythroid 2 (NF-E2). The NRF2 transcription factor is ubiquitously expressed, particularly in organs involved in cellular detoxification, such as the liver and kidney [34].

The human *NFE2LE* gene, which is approximately 34 kb and encodes the NRF2 protein, resides in chromosome 2. The human NRF2 protein possesses a basic leucine zipper (bZIP) domain and a glutamate- and aspartate-rich region in the N-terminus segment. The bZIP and the adjacent N-terminal regions share similarities with several other proteins, including NF-E2 and the cap and collar protein (CNC) of drosophila [35]. Thus, NRF2 is thought to belong to the CNC subfamily of transcription factors, which also encompasses NRF1, NRF3, and p45-NF-E2 proteins.

### 2.2. Domains and Interactions

The human NRF2 protein comprises 605 amino acids and is predicted to have a molecular weight of approximately 65 kDa. However, due to the presence of numerous acidic residues in its sequence, the protein is typically detected at around 95–110 kDa in Western Blot analyses [33,36,37]. The NRF2 protein contains seven highly conserved regions referred to as NRF2–ECH homology (Neh) domains, and their roles and interactions are described below and in Figure 1.

The Neh1 domain encompasses the CNC and bZIP regions. The bZIP region (amino acids 525–566) plays a pivotal role in DNA binding and facilitates dimerization with various transcription factors, including MAF proteins, c-Jun, Sp1, and JDP2 [38]. Furthermore, NRF2 contains six lysine residues susceptible to acetylation by p300/CBP (CREB-binding protein) acetyltransferase, which is crucial for DNA binding to the antioxidant response element (ARE)/electrophile response element (EpRE) sequence, characterized as (A/G)TGA(C/T)NNNGC(A/G) [39]. The regulation of target gene expression depends on the type of coactivator that NRF2 interacts with. The BTB domain and CNC homolog 1 (BACH1) protein also plays a role in the expression of genes controlled by ARE sequences under basal conditions and can form dimers with small-MAF factors and obstruct the NRF2 transcriptional activation, as exemplified in the case of genes such as NADP(H): dehydrogenase quinone 1 (NQO1) and heme oxygenase 1 (HO-1) [40].

This domain also contains a nuclear location site (NLS) [41], which facilitates NRF2 translocation into the nucleus. Conversely, NRF2 can exit the nucleus through interaction with the nuclear export signal (NES) sequence and the chromosomal maintenance 1 (CRM1) protein [42]. Additionally, phosphorylation of the S550 residue by AMP-activated protein kinase (AMPK) plays a crucial role in the NRF2 nuclear translocation [43]. Finally, the K533 residue is significant in the SUMOylation process, mediated by the UBC9 protein, leading to NRF2 stabilization [44].

Neh2 is one of the degron domains essential for regulating NRF2 protein degradation. KEAP1, along with other ubiquitin ligases such as CR6-interacting factor 1 (CRIF1) and WD repeat domain 23 (WDR23), interacts with NRF2 via this domain [45,46,47]. Two specific sequences, ETGE (amino acids 79–82) and DLG (amino acids 29–31), engage with KEAP1 [47]. The NRF2/KEAP1 binding model is referred to as “hinge-and-latch”, with the hinge signifying a robust bond between the ETGE motif of NRF2 and a Kelch domain of KEAP1, while the latch represents a weaker bond between the DLG pattern of NRF2 and another Kelch domain of KEAP1 [48]. In addition to its interaction with NRF2, KEAP1 binds to the actin cytoskeleton, retaining the transcription factor in the cytoplasm [49]. KEAP1 can further engage with other proteins, including the cullin-3 E3-ubiquitin ligase complex (CUL-3). This complex, comprising the CUL-3 protein and ring box 1 protein (RBX1) [50], recruits the E2 enzyme, leading to the ubiquitination of NRF2 and subsequent proteasomal degradation [51,52,53]. Notably, seven lysine residues within the Neh2 domain are pivotal for KEAP1-dependent polyubiquitination [53]. Additionally, the DIDLID motif interacts with WDR23, and this domain possesses an α-helix with seven lysine amino acid residues serving as ubiquitin acceptors [54,55]. Another NLS sequence plays a key role in the NRF2 nuclear translocation under the control of Karyopherin α1 and Karyopherin β1 importins [41]. The S40 residue can undergo phosphorylation by protein kinase C (PKC), facilitating the NRF2 release from KEAP1 [56].

Neh3, the C-terminal domain of NRF2, is essential for transcriptional activation [57]. This region encompasses a third NLS sequence and two lysine residues subject to acetylation [58]. Through this domain, NRF2 recruits the chromo-ATPase/helicase DNA-binding protein 6 (CDH6) co-activator [57].

Neh4 and Neh5 represent the transactivation domains. In this domain, CBP/p300 and Rac family small GTPase 3 (RAC3) interact with NRF2 [59], while the E3 ubiquitin ligase HRD1 also binds to NRF2 [60]. A second redox-sensitive NES sequence (amino acid sequences between 175 and 186) is situated within the Neh5 domain [61].

Neh6 constitutes the second degron domain. Beta-transducin repeat-containing protein (βTrCP) ubiquitin ligase associates with NRF2 through the DSGIS motif, following prior phosphorylation at S344 and S347 by glycogen synthase kinase 3 beta (GSK3β), and the DSAPGS motif. This region allows for the regulation of NRF2 turnover under oxidative stress, though the exact mechanism remains to be fully elucidated [54].

Neh7 domain interacts with retinoid X receptor alpha (RXRα) protein and elicits NRF2 repression [62].

### 2.3. Mechanisms of NRF2 Activation and Inhibition

Under basal conditions, NRF2 forms a tight complex with its repressor, KEAP1, which recruits ubiquitin ligases, ultimately resulting in NRF2 polyubiquitination and subsequent proteasomal degradation [47]. However, during specific types of cellular stress, such as oxidative stress, exposure to heavy metals, or exposure to chemopreventive agents, NRF2 is liberated from its association with KEAP1 [63]. Subsequently, NRF2 translocates to the nucleus to engage and regulate specific target genes that contain ARE sequences [64,65]. The small fraction of NRF2 present in the nucleus is subsequently phosphorylated by the Fyn kinase, and then exported out of the nucleus [66]. As illustrated in Figure 2, the release of NRF2 from KEAP1 and its activation involves a complex interplay of multiple mechanisms.

(1)KEAP1 is the main repressor of NRF2 activation and is rich in redox-sensitive cysteine residues. Notably, Cys151, Cys273, and Cys288 in the KEAP1 protein are involved in the interaction with NRF2 [67]. During oxidative stress, these cysteine residues, functioning as ROS sensors, become oxidized, causing a structural alteration in KEAP1. In the canonical pathway, this oxidative change disrupts the hinge-and-latch complex and severs the connection with the NRF2 DLG pattern. Consequently, the NRF2–KEAP1 complex undergoes a conformational shift that prevents NRF2 ubiquitination [47,55].(2)Other proteins such as p21 or p62 can compete with KEAP1 for binding to the NRF2 DLG motif through the non-canonical pathway. This competitive binding results in a conformational change that hinders NRF2 degradation [68].(3)Different phosphorylation events involving NRF2 can lead to its dissociation from KEAP1. For instance, protein kinase C (PKC) induces phosphorylation of the Ser40 residue [69], which is situated in the Neh2 domain that interacts with KEAP1 [56]. This phosphorylation hinders the binding of NRF2 to KEAP1, preventing its sequestration by KEAP1.(4)The antioxidant iASPP competes with NRF2 for KEAP1 binding via a DLT motif and induces NRF2 activation [70].(5)NRF2 can undergo glycation, rendering it unstable and impairing its binding to small MAF proteins and transcriptional activation. Fructosamine-3-kinase (FN3K) can promote deglycation of the NRF2 protein [71].(6)NRF2 glutarylation, regulated by the mitochondrial protein glutaryl-CoA dehydrogenase (GCDH), enhances protein stability and transcriptional activity [72].

All these mechanisms ultimately contribute to an increase in the NRF2 lifespan, cellular concentration, and translocation to the nucleus, where it functions as a transcription factor.

Conversely, the NLS sequences of NRF2 are sensitive to oxidative stress and phosphorylation, potentially leading to an increase in the export of NRF2 from the cytoplasm to the nucleus [41]. Furthermore, certain transcription factors can compete with NRF2 for ARE sequences in the nucleus. In this context, BACH1, a MAF-related transcriptional repressor, can bind to ARE sequences and act as a negative competitor of NRF2 [73,74]. Antioxidants can induce phosphorylation of BACH1 by an unidentified kinase, causing its exit from the nucleus, protein degradation, and NRF2 activation [75]. Intriguingly, BACH1 is a target gene of NRF2, and its induction creates a negative feedback loop to regulate ARE-mediated genes [76].

### 2.4. Genes Regulated by NRF2

NRF2 regulates the expression of numerous genes, which have been extensively studied [77,78,79]. This non-exhaustive list includes:(1)Antioxidant proteins: NRF2 finely regulates redox homeostasis by controlling the expression of antioxidant enzymes and facilitating the production of glutathione (GSH). Some of these antioxidant proteins include biliverdin reductase B, ferritin, glutamate-cysteine ligase (GCL), superoxide dismutases (SODs), glutathione peroxidases (GPXs), peroxiredoxins (PRXs), and glutathione reductase (GR). Notably, GCL is a critical enzyme involved in the synthesis of the potent antioxidant GSH.(2)NADPH-regenerating enzymes: NRF2 plays a crucial role in governing metabolic reprogramming and the generation of NADPH, which is pivotal in cellular antioxidant systems. Key enzymes in this category include glucose-6-phosphate dehydrogenase (G6PD), malic enzyme 1 (ME1), and 6-phosphogluconate dehydrogenase (6PGD).(3)Cytoprotective proteins: NRF2 regulates important proteins like HO-1 and metallothioneins. HO-1 catalyzes the breakdown of heme, resulting in the production of various compounds, including biliverdin, carbon monoxide, and iron. The cytoprotective effect of HO-1 is mediated indirectly through the generation of biliverdin and the potent antioxidant bilirubin. Notably, HO-1 expression has been observed to be higher in NRF2 knockout K-ras^G12V^ 293T cells compared to wild-type NRF2 cells, suggesting that these enzymes can also be regulated by other transcription factors and signaling pathways [80].(4)Phase 1 enzymes: Phase I metabolism involves the reduction, oxidation, or hydrolysis of molecules such as drugs or toxic compounds. NRF2 regulates a range of enzymes in this category, including alcohol dehydrogenases (ADHs), aldehyde dehydrogenases (ALDHs), cytochromes P450 (CYPs), NQO1, and carboxyl esterase (CES).(5)Phase 2 enzymes: NRF2-dependent conjugation reactions are crucial for the detoxification of various xenobiotics. These reactions are carried out by glutathione S-transferases (GSTs), sulfotransferases (SULTs), and UDP-glucuronosyl transferases (UGTs).(6)Transport proteins: NRF2 also regulates transport proteins such as multi-drug resistance-associated proteins (MRPs) and neutral amino acid transporters through ARE sequences in their promoters. MRPs play a role in drug resistance.(7)Chaperone proteins: Chaperone proteins are responsible for ensuring the proper three-dimensional folding of other proteins, thus facilitating their maturation. NRF2 regulates various chaperone proteins, including heat-shock proteins (HSPs).(8)Transcription factors: NRF2 regulates the expression of MAF proteins, as well as BACH1 and NRF2 itself, through the ARE sequences in their promoters.

### 2.5. NRF2: A Double-Edged Sword

The transcription factor NRF2 exhibits a dual role in cancer. Its impact can vary depending on the conditions, as it may exert a cytoprotective effect or promote tumor progression, chemotherapy resistance, and metastasis.

#### 2.5.1. NRF2: The Bright Side

Several studies utilizing NRF2 knockout mice have provided valuable insights into the role of NRF2. These studies reveal two significant findings: Firstly, NRF2 knockout mice exhibit markedly lower basal levels and inducible expression of detoxification and antioxidant genes, which is observed in various tissues, including the bladder [81], brain [82], and lungs [83]. Secondly, NRF2 knockout mice display increased sensitivity to different forms of stress, including exposure to various xenobiotics and environmental stressors such as cigarette smoke [84].

Additional investigations using liver-specific KEAP1 conditional knockout (KEAP1-cKO) mice have demonstrated that these mice exhibit high resistance to acetaminophen, even at doses that prove lethal for wild-type mice [85]. Acetaminophen is metabolized in vivo to N-acetyl-p-benzoquinone imine (NAPQI), an extremely toxic reactive metabolite. In wild-type mice, NAPQI binds covalently to proteins, leading to adduct formation and severe toxicity. However, in KEAP1-cKO mice, NRF2 activation promotes the synthesis of GSH, which facilitates the detoxification of NAPQI and inhibits the formation of toxic protein adducts [85].

A transient increased ROS level will lead to cell adaptation to stress, a phenomenon known as hormesis, a biphasic dose response characterized by low-dose stimulation and high-dose inhibition. In this context, NRF2 serves as a hormetic mediator [86]. Under normal conditions, KEAP1 will sequester NRF2 leading to its degradation. A moderate increase in ROS will activate NRF2 and trigger antioxidant capacity to restore the redox balance. At the same time, NRF2 will induce BACH1 transcription to generate a negative feedback and restrict it once the ROS levels are stabilized. However, prolonged exposure to ROS may alter the hormetically regulated NRF2 mechanisms and induce cellular redox imbalance that could promote pro-oncogenic effects.

Therefore, NRF2 is considered “beneficial” in the context of a healthy cell, as it induces the expression of cytoprotective genes, enabling the elimination of ROS and detoxification of carcinogens. These effects protect DNA from damage caused by oxidative stress and toxic agents. In addition, NRF2 is directly implicated in various DNA repair pathways resulting in the maintenance of genome integrity and regulation of cell cycle [87]. In this regard, NRF2 activators hold potential as chemopreventive agents. Figure 3 provides a schematic representation of the NRF2 anti-oncogenic functions.

#### 2.5.2. NRF2: The Dark Side

Constitutive activation of NRF2 has been proposed to contribute to chemotherapy resistance in cancer cells due to the increased expression of cytoprotective enzymes [88]. This drug resistance was effectively counteracted by siRNA targeting NRF2 and was observed to be diminished in NRF2 knockout K-ras^G12V^ transformed cells, confirming the protective role of NRF2 in shielding cancer cells from the effects of chemotherapeutic agents [80,89]. Additionally, NRF2 plays a role in modulating tumor metabolism, promoting immune evasion, and participating in cancer metastasis [90,91].

It is worth noting that mutations in both KEAP1 and NRF2 are common in many cancers [77] and are associated with a poor prognosis [92]. The primary mutations affect the binding domains between KEAP1 and NRF2, such as the Kelch domains and DLG and ETGE motifs, respectively. These mutations disrupt the conformation of the NRF2–KEAP1 complex, leading to the accumulation of NRF2 in cancer cells and the overexpression of ARE-mediated genes, including MRP efflux pumps. The increased number of efflux pumps enhances the resistance of tumor cells to chemotherapy by facilitating the export of chemotherapy agents out of cancer cells [77]. Overexpression of HO-1 in various tumors appears to benefit cancer cells by increasing resistance to stress and apoptosis, promoting rapid cancer cell growth, angiogenesis, and metastasis [93]. Chemotherapy-resistant cancer cells are found to have higher levels of GSH compared to non-resistant cancer cells [94]. When GSH levels are reduced, sensitivity to chemotherapy increases due to elevated ROS levels [95].

NRF2 has demonstrated the ability to prevent chemically induced lung cancer but has been shown to accelerate the proliferation of pre-existing tumors [96]. While NRF2 activators are promising for chemoprevention, inhibiting NRF2 could be a viable strategy for anti-cancer therapy. These inhibitors have the potential to suppress tumor proliferation, enhance apoptosis, and increase susceptibility to chemotherapy [97]. However, NRF2-based chemoprevention using sulforaphane was not effective in a KRAS^G12D^-induced lung cancer model [98]. Currently, no specific inhibitor targeting NRF2 has been developed. Since NRF2 belongs to the large family of CNC transcription factors, all of which contain the bZIP sequence, it is necessary to focus on other domains to achieve higher specificity [99]. This will be discussed in more detail in Section 5.

## 3. NRF2 in the Context of Cancer Promotion

### 3.1. Pro-Oncogenic Functions of the KEAP1–NRF2 Pathway

NRF2, by promoting cell survival under stress conditions, can facilitate tumor development through several molecular mechanisms that ultimately protect cancer cells. This aberrant activation provides cancer cells with several advantages over normal cells, such as enhancing their tumorigenic capacity, resistance to therapeutic agents and antioxidant activity, thus leading to what is often referred to as “NRF2 addiction”, which can transform the cellular protective mechanism into a promoter of cancer. Additionally, mutations in NRF2 can result in gain-of-function effects [100]. In response to oxidative stress, increased NRF2 activation provides cytoprotection and maintains higher levels of ROS in cancer cells compared to healthy cells. This heightened ROS generation can lead to DNA point mutations, deletions, and gene amplifications. These genetic alterations may activate proto-oncogenes or inactivate tumor suppressor genes. Oncogenic signaling, such as mutant K-ras, often generates elevated ROS levels [11,16]. Increased NRF2 activation is one of the mechanisms employed by cancer cells to keep these ROS levels below the toxic threshold. NRF2 achieves this by inducing the transcription of antioxidant enzymes like PRXs and GPXs, as well as by generating both NADPH and GSH. Genetic disruption of the NRF2 pathway impairs mutant K-ras-induced proliferation and tumorigenesis, underscoring the crucial role of NRF2 and the antioxidant system in carcinogenesis [101]. Furthermore, the genetic abrogation of NRF2 enhances the sensitivity of K-ras^G12V^ to various pro-oxidant chemotherapeutic agents both in vitro and in vivo [80].

A significant concern in cancer therapy is the development of resistance to both chemotherapy and radiotherapy by cancer cells. For instance, excessive activation of NRF2 through pretreatment with a synthetic antioxidant has been shown to enhance the survival of neuroblastoma cells treated with chemotherapeutic drugs [24]. In radiotherapy, ROS production induced by the treatment leads to a gradual reduction in GSH levels, triggering NRF2 activation and subsequent synthesis of antioxidant enzymes such as GCLC and HO-1 in prostate cancer cells [102].

The role of ROS in cancer is complex. On the one hand, suppressing ROS levels is essential to prevent cancer development, as ROS are implicated in promoting and sustaining carcinogenesis [103,104]. On the other hand, certain drugs can stimulate ROS production and can effectively eliminate cancer cells [105]. Consequently, it is crucial to delineate the specific pathways and balance between ROS and NRF2 to comprehend the paradoxical role of the KEAP1/NRF2 pathway in cancer.

Figure 4 provides a schematic representation of the NRF2 pro-oncogenic functions. In the subsequent sections, we will delve into some of the most pertinent characteristics of cancer cells that are regulated by NRF2 activation.

### 3.2. Role of NRF2 in the Dysregulation of Cell Proliferation

The KEAP1–NRF2 pathway serves as a crucial defense mechanism against oxidative and electrophilic stress, both in normal cells and in cancer cells, albeit with different outcomes [106]. In normal cells, as well as for protection against carcinogenesis, transient activation of NRF2 is essential for survival. However, persistent activation of this pathway can be detrimental, especially in a cancerous context, as NRF2 takes on a pro-tumor role by promoting sustained cancer cell proliferation through various mechanisms [106]. Studies conducted with cell lines derived from lung cancer, pancreatic cancer, and hepatocellular carcinoma have demonstrated that the decoupling of KEAP1–NRF2 alters proliferation dynamics. NRF2-negative cells tend to proliferate more slowly, whereas cells with KEAP1 deletion exhibit faster proliferation rates compared to their counterparts. Mechanistically, NRF2 regulates cancer cell proliferation by transcriptionally activating key proteins such as nephronectin (NPNT), bone morphogenetic protein receptor 1A (BMPR1A), insulin-like growth factor 1 (IGF1), integrin beta chain-2 (ITGB2), platelet-derived growth factor C (PDGFC), vascular endothelial growth factor C (VEGFC), and Jagged 1 protein (JAG1) [79].

In addition to the aforementioned genes, NRF2 governs the expression of genes necessary to meet the continuous protein synthesis demands of cancer cells. These genes include phosphoglycerate dehydrogenase (PHGDH), phosphoserine aminotransferase 1 (PSAT1), phosphoserine phosphatase (PSPH), and serine hydroxymethyltransferase (SHMT), achieved through the activation of activating transcription factor 4 (ATF4) [79,107]. It is important to highlight that ATF4 also regulates the expression of genes involved in oxidative stress, amino acid synthesis, differentiation, metastasis, and angiogenesis. ATF4 expression is frequently elevated in cancer and induced by factors in the tumor microenvironment that drive cancer progression [108]. In summary, excessive activation of NRF2 promotes cancer cell proliferation by inducing cytoprotective genes as well as genes associated with cell proliferation [109]. This heightened activation leads cancer cells to exhibit a “NRF2 addiction” phenotype, characterized by an abnormal accumulation of NRF2 in both murine and human cancers [106]. Consequently, targeting the NRF2 pathway holds promise for suppressing tumor growth, serving as the foundation for drug development against NRF2 [110].

### 3.3. Role of NRF2 in Tumor Metabolism

Cancer cells face challenges such as oxygen stress, nutrient scarcity, and increased oxidative stress, requiring substantial energy to support their proliferation and survival. In contrast to normal differentiated cells that rely on the mitochondrial tricarboxylic acid (TCA) cycle and oxidative phosphorylation (OXPHOS) for energy production, cancer cells prefer aerobic glycolysis with a slower TCA metabolism [111]. While mitochondrial OXPHOS generates 36 molecules of adenosine triphosphate (ATP) from one molecule of glucose, glycolysis produces only two ATP molecules for the same amount of glucose. The preference for aerobic glycolysis in cancer cells is attributed to several factors: (1) rapid glucose-to-lactate conversion, around 100 times faster than the TCA cycle; (2) interconnectivity with various metabolic pathways; (3) advantageous positioning in nutrient and amino acid competition; (4) the ability of lactate to facilitate metabolic symbiosis between cancer cells and neighboring cells in the tumor microenvironment. Additionally, ROS have diverse effects on cancer cell metabolic reprogramming by disrupting key metabolic enzymes and redox-sensitive transcription factors. Furthermore, the induction of mitochondrial dysfunction leads to increased ROS generation, initiating a vicious cycle within cancer cells.

Redox-sensitive signaling pathways, including phosphoinositide 3-kinase (PI3K)/protein kinase B (AKT/PKB)/mechanistic target of rapamycin (mTOR), p53, AMPK, hypoxia-inducible factor 1 alpha (HIF-1α) and NRF2, exert significant influence on cancer cell metabolism [112]. For example, the expression of glucose transporters (GLUTs) and glycolytic enzymes, such as hexokinase, is predominantly regulated by HIF-1α activity [113,114]. NRF2 plays a role in enhancing glutaminolysis [115], nucleotide metabolism [116], and the upregulation of metabolic enzymes within the pentose phosphate pathway (PPP) [117]. This pathway leads to the production of NADPH, a crucial cofactor involved in antioxidant systems (e.g., GSH, thioredoxin) and the maintenance of ROS levels below toxic thresholds. Additionally, increased activation of NRF2 in cancer cells results in elevated expression of enzymes involved in intermediate metabolism, such as transketolase (TKT) and G6PD. These enzymes counteract oxidative stress and contribute to metabolic reprogramming and cell proliferation [115]. Several studies have provided evidence for the indirect role of NRF2 in cancer cell proliferation by its regulation of specific non-coding microRNAs (miRNAs), including miR-1 and miR-206, which can induce translational repression. These miRNAs inhibit TKDT and G6PD genes, and their repression by histone deacetylase 4 (HDAC4) through NRF2 supports cancer cell growth [118]. The ability of miRNA to regulate several genes indicates they are “master regulators” of important biological processes.

NRF2 also plays a pivotal role in regulating genes essential for the synthesis of nicotinamide adenine dinucleotide phosphate (NADPH) [117] and GSH [119], two molecules that are critically involved in cell proliferation. Furthermore, in addition to its influence on glucose metabolism, NRF2 exerts negative regulation on genes associated with fatty acid and lipid metabolism [120]. It is important to note that while cancer cell metabolism tends to favor aerobic glycolysis over mitochondrial respiration, this does not imply a complete shutdown of mitochondrial function; mitochondria continue to have significance in energy metabolism and tumor progression. A notable example of this mitochondrial importance is the glutamine addiction observed in certain cancer cells [121]. In KRAS mutant lung cancer cells, the cooperative actions of LKB1 and NRF2 drive metabolic reprogramming and induce a reliance on glutamine [122]. Glutamine can be converted into glutamate and alpha-ketoglutarate (αKG) within cells to fuel the TCA cycle in mitochondria. Additionally, glutamate can be exchanged with cystine through the xCT transporter, a process regulated by NRF2 [123]. This exchange contributes to the generation of GSH [124] and facilitates nucleotide and amino acid synthesis [125].

Mitochondria are often regarded as a major source of ROS in cancer cells, primarily due to the release of superoxide anions and hydrogen peroxide by respiratory complexes and NADPH oxidases (NOXs). Elevated ROS production can lead to rapid depolarization of mitochondrial membrane potential and impairment of OXPHOS complexes. Consequently, damaged mitochondria produce even more superoxide anions and hydrogen peroxide, creating a vicious cycle of ROS propagation and NRF2 activation in cancer cells [126]. In certain scenarios, such as the loss of PKC in hepatocytes, this can induce autophagy and enhance OXPHOS, resulting in increased ROS generation and NRF2 activation, ultimately promoting liver tumorigenesis [127]. Interestingly, recent research has revealed that the activation of NRF2, achieved by inhibiting KEAP1, can reduce the viability of several lung cancer cell lines, suggesting that high NRF2 activity may lead to cell death and sensitize cancer cells to chemotherapeutic agents [128]. NRF2 induction contributes to NADH-dependent reductive stress through the upregulation of ALDH3A1. Furthermore, cells treated with KEAP1 inhibitors or harboring KEAP1 mutations exhibit selective vulnerability to mitochondrial complex I inhibitors, resulting in NADH oxidation impairment and reductive stress [128].

### 3.4. Role of NRF2 in Cell Death

ROS can induce damage to various macromolecules and trigger cell death. Activation of NRF2 in cancer cells serves to prevent such injuries and cell death. Some chemotherapeutic agents can generate ROS and induce DNA damage, and NRF2 has been demonstrated to confer drug resistance through various mechanisms [129]. These mechanisms include enhanced GSH synthesis, increased expression of antioxidant enzymes, modulation of molecules involved in drug metabolism, and regulation of transporters.

Emerging evidence indicates that NRF2 plays a crucial role in determining cell death fate. The inhibition of NRF2 via siRNA resulted in reduced expression of B-cell lymphoma-2 (BCL2) in human liver cancer cells, enhancing apoptosis induced by etoposide [130]. Conversely, NRF2 mediates the activation of apoptotic signaling in melanoma by modulating the activity of GCDH [72]. NRF2 also confers resistance to ferroptosis, an iron and ROS-dependent form of cell death characterized by the accumulation of lipid peroxides. NRF2 target genes, such as GPX4 and xCT, play roles in preventing lipid peroxidation and ferroptosis [131]. NRF2 also plays a pivotal role in inhibiting inflammasomes and protecting cells against pyroptosis, a form of inflammatory cell death. Activation of inflammasome complexes involves caspases, leading to the release of cytokines like interleukin-1 (IL-1) and the cleavage of gasdermins, triggering pyroptosis. ROS are involved in many steps of IL-1 signaling, inducing inflammation and the infiltration of immunosuppressive cells into the tumor environment. Calpain expression and enzyme activity can be regulated in a redox-dependent manner, mainly by modulating intracellular concentrations of free ionized calcium [132,133]. ROS can also activate the NLRP3 inflammasome complex and caspase-1 [134]. Conversely, NRF2 has the ability to suppress ROS levels in mouse skin tissues and inhibit the NLRP3 inflammasome [135]. Interestingly, inflammasome activation leads to rapid degradation of NRF2 by an as-yet-unknown mechanism [136].

There is substantial evidence demonstrating a crosstalk between NRF2 and autophagy. Silencing NRF2 promotes autophagy in pancreatic cancer cells in response to ROS stress [137]. Upon autophagy induction, the p62 protein accumulates and activates NRF2 by interacting with KEAP1 [138]. Notably, p62 is itself a target of NRF2, creating a positive feedback loop [139]. P62 serves as a central protein in various signaling pathways, interacting with ubiquitinated proteins and acting as an autophagy receptor for protein and mitochondrial degradation. In processes involving oxidative stress, both dependent and independent mechanisms disrupt the autophagy pathway, leading to increased p62 levels and cytoplasmic accumulation of KEAP1 in inclusion bodies, resulting in NRF2 activation upon autophagy-related genes Atg7 and Atg5 depletion [138]. In this context, the KEAP1 interaction with p62 induces autophagy, leading to KEAP1 degradation, subsequent NRF2 stabilization, and activation in MEF and HEK293 cells [140].

## 4. Role of NRF2 and ROS on Some Critical Cellular Processes

### 4.1. NRF2 and ROS in Tumor Immunology

ROS and NRF2 play significant roles in modulating and, at times, dysregulating immune responses in both healthy and cancerous tissues. NRF2 is a key regulator of anti-inflammatory gene expression [141], while ROS can trigger extensive inflammation in the tumor microenvironment [142].

ROS are essential for the biology and activation of various immune cell types. During infections, one of the initial defense mechanisms involves the activation of NOX enzymes in endosomal compartments, resulting in the production of superoxide anions (O_2_^•−^) by phagocytes. Superoxide anions are subsequently metabolized by enzymes like SOD and myeloperoxidase to generate hydrogen peroxide and hypochlorous acid (HOCl), which are used to eliminate pathogens within phagolysosomes [143]. ROS play critical roles in stimulating various immune cell types, including dendritic cells [144], natural killer (NK) cells [145], T cells [146,147], and B cells [148], by affecting metabolism, redox-sensitive signaling pathways, and cytokine secretion, among other functions. However, excessive ROS levels, especially when originating from external sources, can have negative effects on immune cell functions in pathological conditions and cancer. Elevated ROS levels and alterations in antioxidant capacity can disrupt immune cell functions, influencing their sensitivity to chemotherapeutic agents and immunotherapy. Excessive ROS in the extracellular environment can lead to T cell death [149] or dysfunction [150] and select immunosuppressive cells that possess stronger antioxidant defenses [151,152,153].

Comparatively, NRF2 can regulate various immune responses. It can inhibit redox-sensitive signaling pathways, such as NF-κB, and the production of cytokines, thus playing a role in dampening initiated immune responses [154]. For example, the activation of NRF2 by tert-butyl hydroquinone can suppress the production of interferon-gamma (IFN-γ) and promote the production of Th2 cytokines in mouse T cells [155]. However, the effect of NRF2 on immune responses can vary depending on the context. In human T lymphocytes, for instance, neither inhibition nor activation of NRF2 appears to significantly alter the expression of inflammatory molecules like IFN-γ, IL-2, and TNF-α [156]. Deficiency of KEAP1, which leads to NRF2 activation, can have implications for immune cell development and homeostasis. In particular, it can disrupt the development and homeostasis of natural killer T (NKT) cells by altering cellular metabolism [157]. NRF2 activation can also impact the metabolism and expression of forkhead box P3 (FoxP3) in regulatory T cells, potentially diminishing peripheral tolerance [158].

NRF2 activation in cancer cells can influence the polarization of macrophages toward M2-like populations, which are associated with tumor progression [159]. In tumor-bearing mice, NRF2 helps protect immunosuppressive myeloid-derived suppressor cells (MDSCs) from high ROS levels in the tumor microenvironment [151], which contributes to the creation of a hostile environment for T cells. As a result, NRF2 knockout (KO) mice tend to develop lung tumors earlier than wild-type mice, characterized by a lower number of intratumoral T cells and increased infiltration of macrophages and MDSCs [160].

ROS have the capacity to increase the expression of programmed death-ligand 1 (PD-L1) in cancer cells as a means of evading the immune system [19]. The expression of PD-L1 was significantly higher in NRF2 knockout (KO) cells, specifically in the 293T T-Rex/K-ras^G12V^ cells, compared to wild-type NRF2 cells. This observation is in line with the role of ROS in promoting PD-L1 expression. Importantly, these NRF2 knockout cells exhibit compromised antioxidant capacity and elevated ROS levels [80]. However, it is essential to note that the NRF2 impact on PD-L1 expression is indirect and mediated through the regulation of ROS levels. NRF2 itself does not directly regulate PD-L1 expression. Additionally, patients with lung squamous cell carcinoma who exhibit NRF2 activation have shown limited responses to PD-L1 immunotherapy [161].

Modulating ROS and NRF2 activity requires caution due to its potential to affect T cell functions. Metformin can enhance CD8^+^ T cell function by upregulating NRF2 and synergizing with PD-1 immunotherapy, while NRF2 deletion abolishes the metformin-mediated antitumor effect [162]. NRF2 activation can restore NK cell metabolism and function in the tumor environment, enhancing anti-tumor activity [163]. The NRF2 activator sulforaphane induces ROS generation, GSH depletion, and secretion of Th17 cytokines in human T cells [164], potentially interfering with the therapeutic efficacy of immune checkpoint antibodies or chimeric antigen receptor (CAR) T cells [165]. Conversely, pre-incubation of human melanoma patient-derived tumor-infiltrating lymphocytes, healthy donor-derived NK cells, and CD19-directed CAR T cells with the NRF2 activator auranofin enhance the efficacy of adoptive cell therapy [166]. These findings underscore the complex and ambiguous roles of ROS and NRF2 in immune cells and the tumor microenvironment. A better understanding of the mechanisms governed by ROS and NRF2 that regulate immune cell functions in tumors is essential for developing novel antitumor strategies. In addition, the effects of NRF2 activators and inhibitors on immune cell functions and the efficacy of current immunotherapies require further investigation.

### 4.2. Interplays between NRF2, ROS and LncRNAs in Cancer

Long non-coding RNAs (LncRNAs) represent a prominent focus in contemporary cancer research, as it has the potential to produce aberrant peptide products that could serve as tumor antigens [167]. Intriguingly, recent findings have highlighted differential LncRNAs transcription responses to H_2_O_2_ between normal and cancer cells, potentially elucidating variations in sensitivity to ROS between these cell types [168]. A significant portion of the differentially expressed LncRNAs are of the extronic type, and prominent transcription factors such as NRF2, HIF-1α, ataxia telangiectasia mutated (ATM), NF-κB, and p53 have been predicted as regulators of the upregulated LncRNAs. This implies that these transcription factors not only govern the expression of protein-coding mRNAs, but also exert control over LncRNAs expression [168].

Various LncRNAs have been shown to be regulated by NRF2 signaling in cancer [169]. For example, smoke and cancer-associated NRF2 signaling in cancer has been associated with the regulation of various LncRNAs [169]. For instance, the smoke and cancer-associated LncRNA-1 (SCAL1) is linked to smoke exposure and cancer, shielding cancer cells from oxidative stress and the harmful effects of cigarette smoke extract [169]. Similarly, lung cancer-associated transcript 1 (NLUCAT1), another upregulated LncRNA in lung cancer patients, confers resistance to chemotherapy and reduces oxidative stress [170]. Several other NRF2-regulated LncRNAs have also been identified, which promote cancer cell proliferation, migration, and resistance to chemotherapy [167].

LncRNAs can function as either positive or negative regulators of NRF2 activity. For example, the metallothionein 1D pseudogene (MT1DP) LncRNA can downregulate NRF2 by stabilizing miR-365a-3p, rendering lung cancer cells more susceptible to ferroptosis inducers [171]. Overall, the precise effects of these LncRNAs on NRF2 activity, oxidative stress, and their broader biological implications in cancer remain largely uncharted territory.

### 4.3. Crosstalk between NRF2 Activation and Phase Separation

ROS have recently been linked to a phenomenon known as phase separation, wherein biomolecules segregate into dense and dilute phases, leading to significant differences in their concentration, mobility, and function [172].

In this context, the p62/sequestosome-1 (SQSTM1) complex undergoes phase separation, forming membrane-less organelles referred to as p62 bodies, which activate the autophagy process [173]. During oxidative stress, p62 bodies can sequester KEAP1, resulting in the liberation of NRF2 [173]. The death domain-associated protein (DAXX) and the autophagy receptor neighbor of *BRCA1* gene 1 (NBR1) promote the oligomerization of p62 and its condensation into a liquid phase, facilitating the NRF2-mediated antioxidant response [174,175,176]. The formation of p62 bodies through liquid–liquid phase separation (LLPS) is regulated by unc-51-like kinase 1 (ULK1)-dependent phosphorylation of p62, which retains KEAP1 and activates NRF2 [177]. Conversely, the modulator of apoptosis 1 (MOAP-1) and the speckle-type BTB/POZ protein (SPOP) act as a negative regulator of NRF2 activation by disrupting the liquid phase condensation of p62 and SQSTM1 bodies [178,179]. These findings highlight the role of p62-KEAP1 liquid phase separation in promoting NRF2 activation under oxidative stress conditions. Moreover, NRF2 can induce enhancer remodeling through a phase separation mechanism, enhancing the transcription of target genes [180].

NRF2 can also indirectly influence the phase separation process. For instance, under glucose deprivation stress in liver cancer cells, NRF2 regulates the expression of sestrin 2 (SESN2). SESN2 disrupts the stability of hexokinase 2 (HK2) mRNA by inhibiting the formation of HK2 mRNA-based LLPS droplets, thereby reshaping tumor metabolism [181].

### 4.4. Role of NRF2 and BACH1 in Cancer Stem Cells and Metastasis

After chemotherapy, certain subpopulations of cancer cells can survive and have been shown to acquire characteristics of cancer stem cells (CSCs). CSCs are distinguished by robust antioxidant defenses, including the thioredoxin and GSH pathways, which confer increased survival potential under conditions of oxidative stress [182]. Additionally, CSCs demonstrate resistance to chemotherapy and acquire an invasive phenotype through the expression of ABC transporters and activation of signaling pathways such as NOTCH, Sonic Hedgehog, β-catenin, and Tafazzin [183]. Therefore, modulating NRF2 activity could represent a promising approach for targeting CSCs, inhibiting metastasis, and overcoming drug resistance. For example, the NRF2 inhibitor brusatol has demonstrated the capability to suppress metastasis and enhance sensitivity to chemotherapy in various animal models of cancer [184].

In a recent study, Cheung et al. reported the pivotal role of temporal and dynamic regulation of ROS levels in supporting the progression of pancreatic cancer [185]. During the initial stages of tumor development, cancer cells generate high levels of ROS and activate various antioxidant proteins, including p53-induced glycolysis regulatory phosphatase (TIGAR) and NRF2, to ensure their survival. Subsequently, as this adaptive phase progresses, TIGAR expression diminishes, resulting in increased ROS levels that contribute to DNA instability. The most notable reduction in TIGAR expression is observed in invasive tumor cells. Loss of TIGAR or NRF2 antioxidant proteins delays the initiation of tumors but elevates ROS levels in pancreatic cancer cells, facilitating a phenotypic shift that enhances invasion and metastatic potential. Notably, treatment with the antioxidant N-acetyl cysteine (NAC) can suppress metastases in TIGAR-deficient tumors. Hence, while targeting antioxidant proteins may offer benefits in managing primary tumors, it could paradoxically enhance tumor cell invasion and metastasis [185].

Conversely, in a KRAS-driven lung cancer model, long-term supplementation with NAC and vitamin E has been demonstrated to promote metastasis and has been attributed to the stabilization of BACH1 and a metabolic shift in invasive cancer cells [186]. Another study using a lung cancer metastasis model showed that NRF2 activation leads to the stabilization of BACH1 and HO-1, both playing pivotal roles in promoting metastasis [187]. These results are consistent with findings indicating that the induction of HO-1 in tumor-associated macrophages can create a premetastatic environment and induce immunosuppression [188]. Additionally, BACH1 has been implicated in the development of an invasive phenotype and pancreatic cancer metastasis [189].

Altogether, these findings indicate that antioxidants might have opposite effects on the occurrence of metastasis, and the therapeutic outcomes with such compounds may depend on the intrinsic antioxidant capacity of invasive cancer cells because metastasis could be promoted by reductive or oxidative stress. The molecular mechanisms by which ROS, NRF2, and BACH1 induce metastatic processes are not fully understood and still need further investigation.

## 5. NRF2 in Cancer Prevention and Its Therapeutic Implications

The KEAP1–NRF2 pathway has been a focal point of extensive research, previously aimed at assessing its potential involvement in chronic human diseases characterized by disruptions in redox balance, including conditions such as diabetes, cardiovascular disease, neurodegenerative diseases, and cancer. In particular, investigations into the chemopreventive properties of natural and synthetic compounds that act as NRF2 activators or KEAP1 inhibitors have been conducted in numerous in vitro and in vivo studies. The elucidation of the molecular mechanisms governing NRF2 regulation has renewed interest in both fundamental and clinical cancer research [190].

Given that NRF2 can exhibit both oncogenic and tumor-suppressive functions, the development of therapeutic strategies based on NRF2 modulation necessitates a meticulous evaluation of the specific context in which its activation occurs. Factors such as tumor histotype, stage, genetic background, therapeutic administration protocols, and the target patient population can all influence the potential success of such treatments.

Numerous NRF2 activators have been identified, although only a select few are currently in clinical development for cancer patients, and some of these may exhibit off-target effects [191,192]. In contrast, the field of NRF2 inhibitors, which could hold substantial promise for cancer therapy, is less advanced. Several clinical trials involving NRF2 inhibitors are currently underway, but many of these drugs lack specificity, possess limited anticancer potency, and carry a significant risk of toxicity [193]. It is noteworthy that radiotherapy and certain chemotherapeutic drugs, such as cisplatin, can induce the generation of ROS, which in turn could activate NRF2 in cancer cells [194,195].

In the following sections, we provide a brief overview of NRF2 modulators, their effects, and their inherent limitations (*cfr* Table 1 for a unified overview).

### 5.1. NRF2 Activators

As previously discussed, the activation of the NRF2 system is a multifaceted process that involves both canonical and non-canonical pathways, triggered by oxidative or electrophilic stresses and mediated by SQSTM1/p62, respectively.

Given the dual nature of the KEAP1–NRF2 protein–protein interaction, extensive research efforts have been dedicated to identifying specific NRF2 activators and/or KEAP1 inhibitors as potential modulators of inflammation and protectors against oxidative stress and carcinogenesis [62,215,216,217]. While NRF2 activation has been regarded as an intriguing therapeutic approach to enhance cellular defenses against external threats, it is important to acknowledge that such activation can also promote pro-oncogenic signaling, contingent upon the context in which NRF2 is activated [91,104,213]. In light of these considerations, the therapeutic utility of NRF2 modulators in cancer treatment has been explored.

The KEAP1–NRF2 pathway can sense electrophiles as potential stressors, which has led to the exploration of electrophilic drugs as a rational approach to induce its activation [218,219]. However, concerns about potential side effects associated with the use of electrophilic compounds have prompted the development of alternative “modulators” of NRF2 activity. Ideally, an optimal NRF2 modulator would not be a potent activator of NRF2, as the degree of activation is directly proportional to its electrophilic nature [220]. Below are examples of both electrophilic and non-electrophilic NRF2 activators.

One of the most well-known and frequently used NRF2 activators in experimental research is sulforaphane. Glucosinolates, which are found in broccoli, undergo hydrolysis to produce sulforaphane, an electrophilic compound known for its ability to activate NRF2 [196,197]. Sulforaphane induces modifications in the cysteine residues of Keap1, disrupting the interaction between KEAP1 and NRF2, ultimately leading to NRF2 activation.

Another important class of chemicals known for their ability to activate NRF2 by inhibiting KEAP1 comprises triterpenoids. These organic compounds are derived from the 5-carbon hydrocarbon isoprene (2-methylbuta-1,3-diene). Triterpenoids can bind to KEAP1, inducing a conformational change that prevents its association with NRF2, ultimately leading to the transactivation of NRF2 target genes. Among these triterpenoids, 2-cyano-3,12-dioxoolean-1,9(11)-diene-28-oic acid (CDDO) is a synthetic derivative of oleanolic acid, and it is a potent NRF2 activator even at nanomolar concentrations. KEAP1 contains 15 cysteine residues that are susceptible to modification by electrophilic compounds. Interestingly, each electrophile targets a unique set of these cysteine residues, a phenomenon referred to as the “cysteine code” [99]. One key cysteine involved in the binding of triterpenoids to KEAP1 is the C151 residue. CDDO has demonstrated antitumor effects in various mouse models of cancer [198,199,213]. To enhance potency, specificity and minimize potential side effects, methyl ester (CDDO-Me) and imidazole (CDDO-Im) derivatives of CDDO have been developed. These derivatives have shown promising results in preclinical studies due to their ability to activate NRF2 at low doses [199].

Conversely, recent studies have explored the therapeutic potential of non-electrophilic compounds as NRF2 activators. Given the potential side effects associated with electrophilic compounds, Zhang et al. conducted an extensive analysis of nearly 200 chemicals to identify potential non-electrophilic NRF2 activators [221]. These compounds were classified based on their structures, encompassing various chemical groups such as phenothiazine, tricyclics, trihexyphenidyl, phenyl pyridine, quinoline-8-substituted, tamoxifen-substituted, and hexetidine-substituted compounds. The results revealed that each class of compounds was capable of inducing diverse NRF2-mediated antioxidant gene expressions in MCF-7 cells [221]. These changes varied among the compound classes and the specific target proteins analyzed, but this systematic approach to identifying biologically active non-electrophilic NRF2 activators holds promise for future development.

An example of a non-electrophilic NRF2 activator is RA839, a low molecular weight non-covalent compound that binds to the NRF2-interacting Kelch domain of KEAP1, leading to activation of multiple pathways associated with NRF2 signaling in bone marrow macrophages [200]. The identification and development of novel non-covalent NRF2 activators have faced challenges due to their lower affinity and potency compared to covalent agents. Dimethyl fumarate (DMF) and its metabolite, monomethyl fumarate (MMF), emerged as a potential NRF2 activator with low toxicity and therapeutic potential against neurodegenerative diseases and chronic inflammation. The unknown mechanism involves upregulation of NRF2 dependent genes and transcription inhibition of pro-inflammatory cytokines [201,202,203]. In addition to current chemical detection strategies that focus on their structure and potential use as non-electrophilic NRF2 activators, ongoing studies are exploring other avenues.

### 5.2. NRF2 Inhibitors

Given the increased expression of NRF2 in several cancers and its role in tumor progression, significant research efforts have been directed towards the development of NRF2 inhibitors and the exploration of their potential anticancer effects.

Glucocorticoid receptor ligands such as dexamethasone and clobetasol propionate can inhibit NRF2 by blocking its transcriptional activity or preventing its nuclear translocation. Bexarotene, a retinoic acid receptor-α (RARα) and RXRα agonist, inhibits NRF2 transcriptional activity [204,205,206] by interacting with the Neh7 domain of NRF2 [205] (*cfr*, Figure 1). However, it is important to note that the pharmacological value of this NRF2 inhibition mechanism is constrained by the multiple effects expected through the regulation of these nuclear receptors.

Several naturally occurring compounds have been reported to possess inhibitory effects on NRF2. One such compound is the quassinoid brusatol, extracted from *Brucea javanica*. Brusatol is known to inhibit NRF2 transcription, rendering tumors and cancer cell lines more sensitive to chemotherapy. However, the mechanism of action of brusatol is not highly specific, as it also hinders protein translation, affecting various short-lived proteins in addition to NRF2 [207,208]. Flavonoids like luteolin and wogonin have also been identified as NRF2 inhibitors. They sensitize cells to anticancer drugs by increasing the instability of NRF2 transcription. While some studies have suggested that these compounds may induce NRF2 activation [209,210], their role as NRF2 inhibitors remains a subject of controversy. Other natural compounds, such as the mycotoxin ochratoxin A and the trigonelline coffee alkaloid, prevent the nuclear translocation of NRF2. Indeed, ochratoxin A is an NRF2 inhibitor acting at different levels such as the inhibition of NRF2 translocation, interference with NRF2–DNA binding, prevention of NRF2-dependent transcription due to histone hypo-acetylation and induction of miR-132 depleting NRF2 protein pools [211]. Regarding trigonelline, it decreases NRF2-dependent proteasome activity [212]. In the case of leukemia cells, malabaricon-A, a plant-derived pro-oxidant, effectively inhibits NRF2 transcriptional activity. This inhibition is characterized by reduced levels of HO-1 protein, the accumulation of ROS, and induction of apoptosis [214].

Furthermore, the antioxidant ascorbic acid has been found to sensitize imatinib-resistant cancer cells. It accomplishes this by decreasing the levels of the NRF2/ARE complex, leading to reduced expression of the glutamate-cysteine ligase catalytic subunit and lower GSH levels [222]. However, it is important to note that the selectivity of these compounds for NRF2 inhibition has not been definitively established.

## 6. Conclusions

The role of NRF2 in cancer remains controversial. Some studies suggest that NRF2 may act as a tumor suppressor, inhibiting carcinogenesis. However, NRF2 expression is elevated in many types of tumors and is associated with poor prognosis, as it provides cancer cells with a survival advantage and resistance to chemotherapy and radiotherapy.

In summary, the findings indicate a protective function for NRF2 in preventing and treating early-stage cancer. Nonetheless, increased NRF2 expression in advanced cancer stages and within cancer stem cells facilitates cancer cell adaptation to tumorigenesis and promotes metastasis. Cancer cells that heavily rely on NRF2 become resistant to chemotherapy and radiation therapy. Therefore, it is reasonable to hypothesize that NRF2 inhibitors could sensitize tumor cells to cancer therapies and reduce the occurrence of metastasis. Conversely, NRF2 plays a vital role in stimulating immune responses and selecting immunosuppressive cells within the tumor environment. Targeting NRF2 could potentially impair the functions of intratumoral immune cells and hinder the therapeutic effects of chemotherapy capable of inducing immunogenic cell death (ICD) or immunotherapies. In all cases, the precise mechanisms of action of NRF2 inhibitors and activators are either unknown or non-specific, which means these compounds are still far from being incorporated into standard chemotherapy protocols.

## Figures and Tables

**Figure 1 antioxidants-13-00070-f001:**
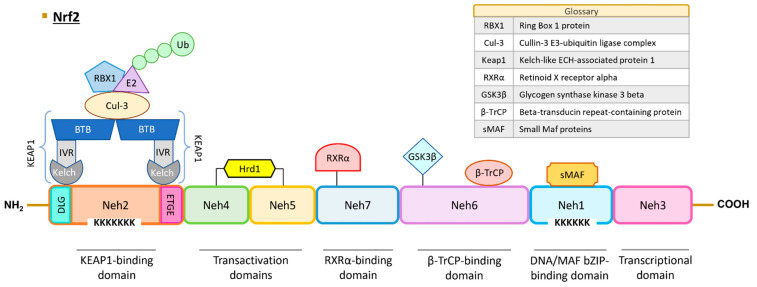
Structure of human NRF2. NRF2 protein contains seven Neh domains with different functions. The Neh1 domain is essential for DNA binding and dimerization with MAF proteins and other transcription factors (c-Jun, Sp-1, and JDP2). The Neh2 domain interacts with KEAP1 through DLG and ETGE motifs, leading to NRF2 ubiquitination and proteasomal degradation. The Neh3, Neh4, and Neh5 are transactivation domains. Neh6 contains a serine-rich region that regulates NRF2 protein stability. The Neh7 domain interacts with RXRα protein and induces NRF2 repression. Abbreviations: BTB, broad complex, tramtrack, and bric-a-brac domain; Cul3, cullin-3; GSK3, glycogen synthase kinase 3 beta; Hrd1, hydroxymethyl glutaryl-coenzyme A reductase degradation protein 1; IVR, intervening region; KEAP1, Kelch-like ECH-associated protein 1; sMAF, small-MAF factors; Neh, NRF2–ECH homology domains; NRF2, nuclear factor erythroid 2-related factor 2; RBX1, ring box 1 protein; RXRα, retinoid X receptor alpha; β-TrCP, beta-transducin repeat-containing protein; Ub, ubiquitin.

**Figure 2 antioxidants-13-00070-f002:**
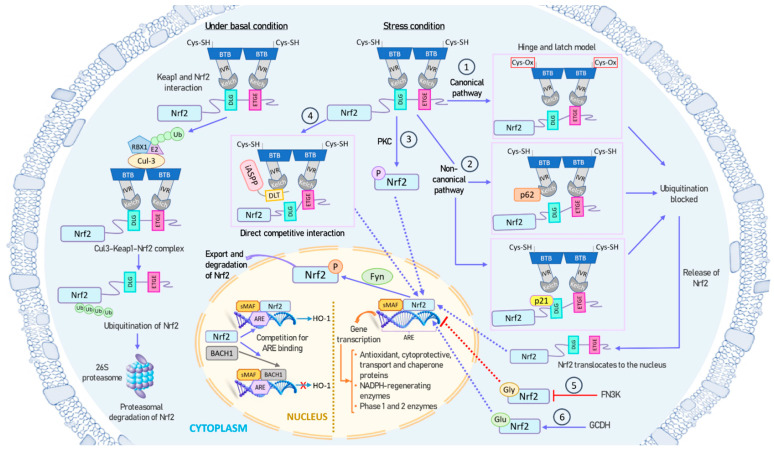
Mechanisms of NRF2 activation and inhibition. Under basal conditions, NRF2 is tightly bound to its repressor KEAP1, which recruits ubiquitin ligases, leading to NRF2 polyubiquitination and subsequent proteasomal degradation (left panel). However, under stress conditions, such as oxidative stress, NRF2 is released from KEAP1 and translocates to the nucleus for binding and regulating specific target genes containing ARE sequences (right and center panels). The small amount of NRF2 present in the nucleus is subsequently phosphorylated by the Fyn kinase and exported out of the nucleus. NRF2 can be activated through both canonical and non-canonical pathways: (1) During oxidative stress, redox-sensitive cysteine residues in KEAP1 are oxidized, causing a structural change in KEAP1. This disturbance in the Hinge-and-latch complex prevents the binding of KEAP1 to the DLG motif of NRF2, thereby preventing NRF2 ubiquitination and degradation. (2) Proteins like p21 or p62 can compete with KEAP1 for binding to the DLG motif of NRF2. (3) PKC phosphorylates the Ser40 residue in NRF2, inhibiting its sequestration by KEAP1. (4) The antioxidant iASPP competes with NRF2 for KEAP1 binding via a DLT motif and induces NRF2 activation. (5) NRF2 can undergo glycation, which can impair its transcriptional activation. FN3K can promote the deglycation of NRF2. (6) GCDH can glutarylate NRF2, increasing its protein stability and transcriptional activity. All these mechanisms ultimately result in an increase in NRF2 lifespan, cellular concentration, and nuclear transport, enabling it to function as a transcription factor. Additionally, BACH1 can compete with NRF2 for binding to ARE sequences in the nucleus. Abbreviations: ARE, antioxidant response element; BACH1, BTB domain and CNC homolog 1; BTB, broad complex, tramtrack, and bric-a-brac domain; Cul3, cullin-3; FN3K, fructosamine-3-kinase; GCDH, glutaryl-CoA dehydrogenase; Gly, glycation; Glu, glutarylation; HO-1, heme oxygenase 1; iASPP, inhibitory member of apoptosis stimulating protein of p53, or ankyrin repeats, SH3 domain and proline-rich region contain protein family; IVR, intervening region; KEAP1, Kelch-like ECH-associated protein 1; sMAF, small-MAF factors; NRF2, nuclear factor erythroid 2-related factor 2; PKC, protein kinase C; RBX1, ring box 1 protein.

**Figure 3 antioxidants-13-00070-f003:**
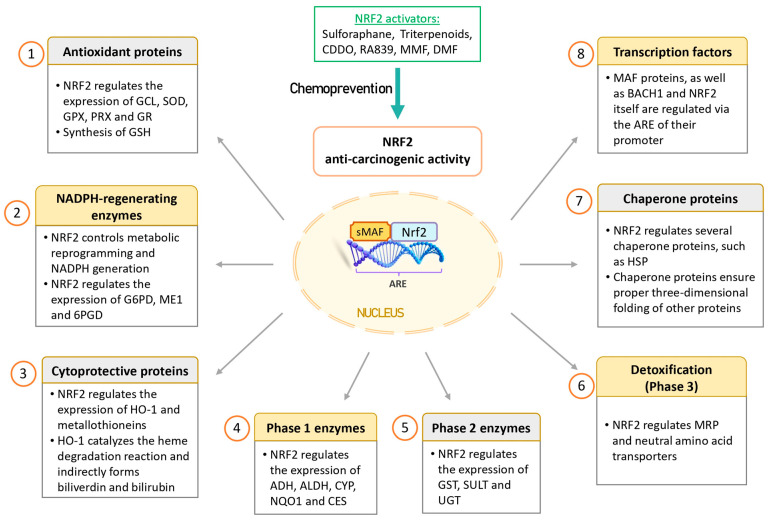
Anti-oncogenic effects of NRF2 in cancer prevention. NRF2 is considered to have anti-oncogenic effects in healthy cells by inducing cytoprotective proteins and enzymes that help eliminate ROS and detoxify carcinogens. This protective mechanism safeguards DNA from damage caused by oxidative stress and toxic agents. Consequently, NRF2 activators may hold potential as chemopreventive agents. Abbreviations: 6PGD, 6-phosphogluconate dehydrogenase; ADHs, alcohol dehydrogenases; ALDHs, aldehyde dehydrogenases; ARE, antioxidant response element; BACH1, BTB domain and CNC homolog 1; CDDO, 2-cyano-3,12-dioxoolean-1,9(11)-diene-28-oic acid; CES, carboxyl esterase; CYPs, cytochromes P450; DMF, dimethyl fumarate; G6PD, glucose-6-phosphate dehydrogenase; GCL, glutamate-cysteine ligase; GPXs, glutathione peroxidases; GR, glutathione reductase; GSH, glutathione; GSTs, glutathione S-transferases; HO-1, heme oxygenase 1; HSPs, heat-shock proteins; ME1, malic enzyme 1; MMF, monomethyl fumarate; MRP, multi-drug resistance-associated protein; NQO1, NADP(H): dehydrogenase quinone 1; NRF2, nuclear factor erythroid 2-related factor 2; PRXs, peroxiredoxins; SODs, superoxide dismutases; SULTs, sulfotransferases; UGTs, UDP-glucuronosyl transferases.

**Figure 4 antioxidants-13-00070-f004:**
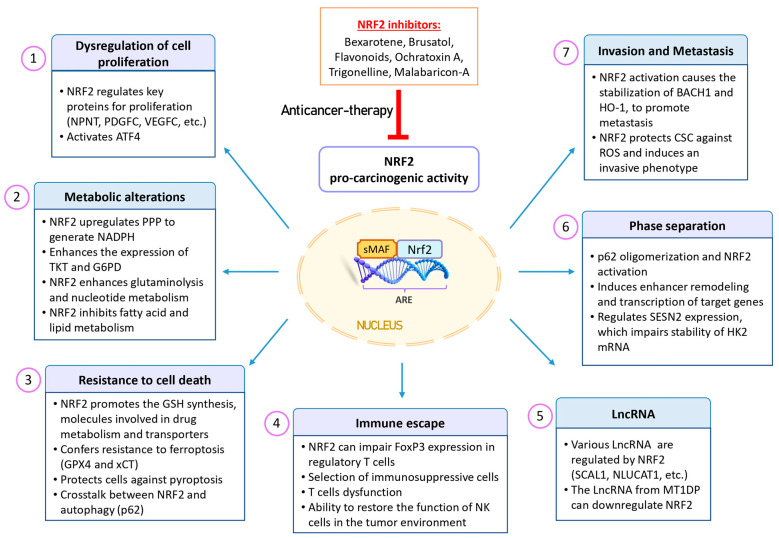
Pro-oncogenic effects of NRF2 in cancer. NRF2 expression is upregulated in various types of tumors and is associated with a poor prognosis. This aberrant activation provides cancer cells with advantages, including increased tumorigenic capacity, resistance to therapeutic agents, and enhanced antioxidant activity. This phenomenon is often referred to as “NRF2 addiction”, where the protective mechanism of NRF2 becomes a driver of cancer growth. Additionally, NRF2 plays a role in multiple processes within cancer cells, including dysregulated cell proliferation, metabolic alterations, resistance to cell death, phase separation, tumor immunology, metastasis, and LncRNA regulation. Given these factors, NRF2 inhibitors may be promising for sensitizing cancer cells to cancer therapies. Abbreviations: ARE, antioxidant response element; ATF4, activating transcription factor 4; BACH1, BTB domain and CNC homolog 1; CSCs, cancer stem cells; FoxP3, forkhead box P3; G6PD, glucose-6-phosphate dehydrogenase; GPX4, glutathione peroxidase 4; GSH, glutathione; HK2, hexokinase 2; HO-1, heme oxygenase 1; LncRNAs, long non-coding RNAs; MT1DP, metallothionein 1D pseudogene; NK, natural killer; NLUCAT1, lung cancer-associated transcript 1; NPNT, nephronectin; NRF2, nuclear factor erythroid 2-related factor 2; PDGFC, platelet-derived growth factor C; PPP, pentose phosphate pathway; ROS, reactive oxygen species; SCAL1, smoke and cancer-associated LncRNA-1; SESN2, sestrin 2; TKT, transketolase; VEGFC, vascular endothelial growth factor C.

**Table 1 antioxidants-13-00070-t001:** Examples of NRF2 modulators and their modes of action.

NRF2 Activators	Modes of Action	References
Sulforaphane	Covalent binding to KEAP1 cysteine residues	[196,197]
Triterpenoids (CDDO)	Target KEAP1 and activation of NRF2 response	[198,199]
RA839	Selective inhibitor of the KEAP1/NRF2 interaction	[200]
MMF/DMF	Activation of NRF2 and upregulation of its target genes	[201,202,203]
**NRF2 Inhibitors**	**Modes of Action**	**References**
Bexarotene	Interaction with the Neh7 domain of NRF2	[204,205,206]
Brusatol	Global protein synthesis inhibitor	[207,208]
Flavonoids	Increase in NRF2 instability	[209,210]
Ochratoxin A	Interference with NRF2 translocation and its DNA binding	[211]
Trigonelline	Reduced nuclear accumulation of the NRF2 protein	[212,213]
Malabaricon-A	Inhibition of NRF2 transcriptional activity	[214]

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
