# Peer review of "The Multifaceted Roles of NRF2 in Cancer: Friend or Foe?"

_antioxidants, 2024, doi:10.3390/antiox13010070_

Round 1
Reviewer 1 Report
Comments and Suggestions for Authors
This is a nice and mostly well written review on the dual role of NRF2 in cancer, covering the beneficial actions of activated NRF2 in healthy cells (with respect to cancer prevention) as well as detrimental actions of dysregulated highly active NRF2 in cancer. I have only a few remarks/suggestions in order to further improve the manuscript.
1) To my mind, the organisation of the manuscript is not really logical. I suggest to include the part on NRF2 activators directly after the discussion of beneficial actions of NRF2 in cancer prevention and also to include here a table with known NRF2 activators and their mode of action. Similarly, I suggest to put the chapter on NRF2 inhibitors after the part on the discussion of negative outcomes of dysregulated NRF2 activity in cancer cells.
2) In regard to the beneficial actions of NRF2 in healthy cells and for cancer prevention, the authors could also discuss hormesis
Minor points:
1) Lines 84/85: Do you mean the gene of the mRNA for NRF2, which has a length of 2.2 kB? Please check!
2) Line 119: Please mention here also the other name of the NRF2 binding site in promoters of its target genes, which refers to its activation by electrophiles: EpRE (Electrophile Response Element).
3) Line 304 (Legend): "Anti‐oncogenic effects of NRF2 in cancer" should read "Anti‐oncogenic effects of NRF2 in cancer prevention"
4) Line 418: "coupling" should read "decoupling"
Author Response
This is a nice and mostly well written review on the dual role of NRF2 in cancer, covering the beneficial actions of activated NRF2 in healthy cells (with respect to cancer prevention) as well as detrimental actions of dysregulated highly active NRF2 in cancer. I have only a few remarks/suggestions in order to further improve the manuscript.
Response: We would like to thank the reviewer for the positive comments, and have carefully considered the suggestions below.
1) To my mind, the organisation of the manuscript is not really logical. I suggest to include the part on NRF2 activators directly after the discussion of beneficial actions of NRF2 in cancer prevention and also to include here a table with known NRF2 activators and their mode of action. Similarly, I suggest to put the chapter on NRF2 inhibitors after the part on the discussion of negative outcomes of dysregulated NRF2 activity in cancer cells.
Response: While we understand the reviewer’s suggestion, we feel that the suggested changes could disrupt the overall presentation flow and create a truncated end. Thus, we would like to keep the original manuscript structure. Regarding the table with known NRF2 activators, we have now included a Table in line 749 summarizing the modes of action of both NRF2 activators and inhibitors.
2) In regard to the beneficial actions of NRF2 in healthy cells and for cancer prevention, the authors could also discuss hormesis.
Response: We thank the reviewer for this suggestion. We have now added few sentences in the section 2.5.1 (lines 298-306) and one reference (reference #86).
Minor points:
1) Lines 84/85: Do you mean the gene of the mRNA for NRF2, which has a length of 2.2 kB? Please check!
Response: Indeed, it was confusing. We have indicated the length of the gene in line 84, which is approximately 34 kb (source: NCBI, Gene ID: 4780).
2) Line 119: Please mention here also the other name of the NRF2 binding site in promoters of its target genes, which refers to its activation by electrophiles: EpRE (Electrophile Response Element).
Response: We have added the EpRE binding sites as suggested in line 119.
3) Line 304 (Legend): "Anti‐oncogenic effects of NRF2 in cancer" should read "Anti‐oncogenic effects of NRF2 in cancer prevention"
Response: As suggested, we have modified the title of the Figure legend in line 316.
4) Line 418: "coupling" should read "decoupling"
Response: We have now corrected the typo in line 431.

Reviewer 2 Report
Comments and Suggestions for Authors
The presented manuscript entitled "The multifaceted roles of NRF2 in cancer: friend or foe?" provides a comprehensive and excellent review of NRF functions in the context of tumorigenesis and cancer therapy. The review is exceptionally extensive and worth publishing.
Nevertheless, I have a few comments that need to be addressed and can make the manuscript even better.
- The NRF2 is implicated in DNA repair. Unfortunately, there is no mention of these functions in the review. Please add.
- NRF3 (lines 75-77). It is a little awkward to describe the normal function of a protein by where it is NOT expressed and how it is involved in pathology. Please modify.
- Line 136. In the description of motifs of NRF2, the authors use numbers in brackets. It's easy to mix up with references. Please add "aa" as indication that authors mean a number of amino acid residues, eg, 23aa-27aa. Or something similar.
- Subsequent numbers in Figure 2 nicely indicate the processes described in paragraph 2.3. I think that this idea should be applied to other figures, too.
- The whole paragraph 2.4 lacks references. Please indicate the proper literature.
- I think that information from Chapter 5 (starting line 710) should also be depicted/summarised in a Figure or a Table.
Author Response
The presented manuscript entitled "The multifaceted roles of NRF2 in cancer: friend or foe?" provides a comprehensive and excellent review of NRF functions in the context of tumorigenesis and cancer therapy. The review is exceptionally extensive and worth publishing.
Nevertheless, I have a few comments that need to be addressed and can make the manuscript even better.
Response: We would like to thank the referee for the constructive comments and suggestions, which are highly valuable and helpful for us to strengthen our manuscript.
1. The NRF2 is implicated in DNA repair. Unfortunately, there is no mention of these functions in the review. Please add.
Response: NRF2 is also involved in DNA repair. We have now added a sentence in the section 2.5.1 (lines 310-312) and one reference (reference #87).
2. NRF3 (lines 75-77). It is a little awkward to describe the normal function of a protein by where it is NOT expressed and how it is involved in pathology. Please modify.
Response: We agree that the sentence was confusing, and have modified it (lines 75-77). Unlike NRF2, the protein NRF3 is not detected in organs involved in detoxification but it is detected in other tissues such as cornea and skin.
3. Line 136. In the description of motifs of NRF2, the authors use numbers in brackets. It's easy to mix up with references. Please add "aa" as indication that authors mean a number of amino acid residues, eg, 23aa-27aa. Or something similar.
Response: We agree with the reviewer’s comment. We have added “amino acids” before the numbers in line 137.
4. Subsequent numbers in Figure 2 nicely indicate the processes described in paragraph 2.3. I think that this idea should be applied to other figures, too.
Response: We have modified the Figures 3 and 4 as suggested by the reviewer.
5. The whole paragraph 2.4 lacks references. Please indicate the proper literature.
Response: We thank the reviewer’s comment, but we have a different point of view. Indeed, as we have explained in the first sentence of the section 2.4 (highlighted in yellow; lines 242-243), previous review manuscripts (references 77-79) have extensively described the genes regulated by NRF2. As the referee pointed out in her/his first comment, the aim of our paper was focused on tumorigenesis and cancer therapy. To our understanding, the inclusion and discussion of more than 30 new references is out of the scope of this review.
6. I think that information from Chapter 5 (starting line 710) should also be depicted/summarised in a Figure or a Table.
Response: This is an interesting suggestion. In the revised version we have now included a Table in line 749 summarizing the modes of action of both NRF2 activators and inhibitors.

Round 2
Reviewer 1 Report
Comments and Suggestions for Authors
The manuscript is fine, I have no further comments.